# Experiences using the poststroke checklist in Sweden with a focus on feasibility and relevance: a mixed-method design

Emma K Kjörk, Gunnel Carlsson, Katharina S Sunnerhagen, Åsa Lundgren-Nilsson

Department of Clinical Neuroscience, Sahlgrenska Academy, Institute of Neuroscience and Physiology, Gothenburg

**Correspondence to**
Mrs Emma K Kjörk;
emma.kjork@neuro.gu.se

## ABSTRACT

**Objective** The wide range of outcomes after stroke emphasises the need for comprehensive long-term follow-up. The aim was to evaluate how people with stroke and health professionals (HPs) perceive the use of the poststroke checklist (PSC), with a focus on feasibility and relevance.

**Design** An exploratory design with a mix of qualitative and quantitative methods.

**Setting** Outpatient care at a university hospital and primary care centres in western Sweden.

**Participants** Forty-six consecutive patients (median age, 70; range, 41–85; 13 women) and 10 health professionals (median age 46; range, 35–63; 7 women).

**Results** Most patients (87%) had one or more problems identified by the PSC. The most common problem areas were life after stroke (61%), cognition (56%), mood (41%) and activities of daily living (39%). Three organisational themes emerged from the focus group discussions. The perception of *the content and relevance of* the PSC was that common poststroke problems were covered but that unmet needs still could be missed. Identifying needs was facilitated when using the *PSC as a tool for dialogue.* The dialogue between the patient and HP as well as HPs stroke expertise was perceived as important. The PSC was seen as *a systematic routine and a base for egalitarian follow-up*, but participants stressed consideration given to each individual. Addressing identified needs and meeting patient expectations were described as challenging given available healthcare services.

**Conclusions** The PSC is a feasible and relevant tool to support egalitarian follow-up and identify patients who could benefit from targeted poststroke interventions. Stroke expertise, room for dialogue and caring for identified needs emerged as important issues to consider when using the PSC. Nutrition, sexuality and fatigue were areas mentioned that might need to be addressed within the discussions. The PSC can facilitate patients in expressing their needs, enhancing their ability to participate in decision-making.

## INTRODUCTION

Poststroke impairments often have long-term negative consequences for social relationships, dependence in daily life and quality of life.[1] Perceived unfulfilled rehabilitation needs and changes in functioning during the first year after stroke[2 3] indicate the need for systematic long-term follow-up. Furthermore, a subtle decline in cognition as well as emotional problems[4 5] can easily be overlooked, leading to difficulty in accessing healthcare services.[6 7]

The adaptation process after stroke is long. In later phases, the focus changes from the stroke itself to more familiar aspects of daily life,[8] which may not be sufficiently targeted in current practice. Concordance is poor between perceived problems and problems detected by standardised assessments. Accordingly, dialogue should complement assessments to ensure that health services are based on patient needs,[9] in line with a person-centred approach.[10]

The poststroke checklist (PSC)[11] was developed to identify long-term care needs

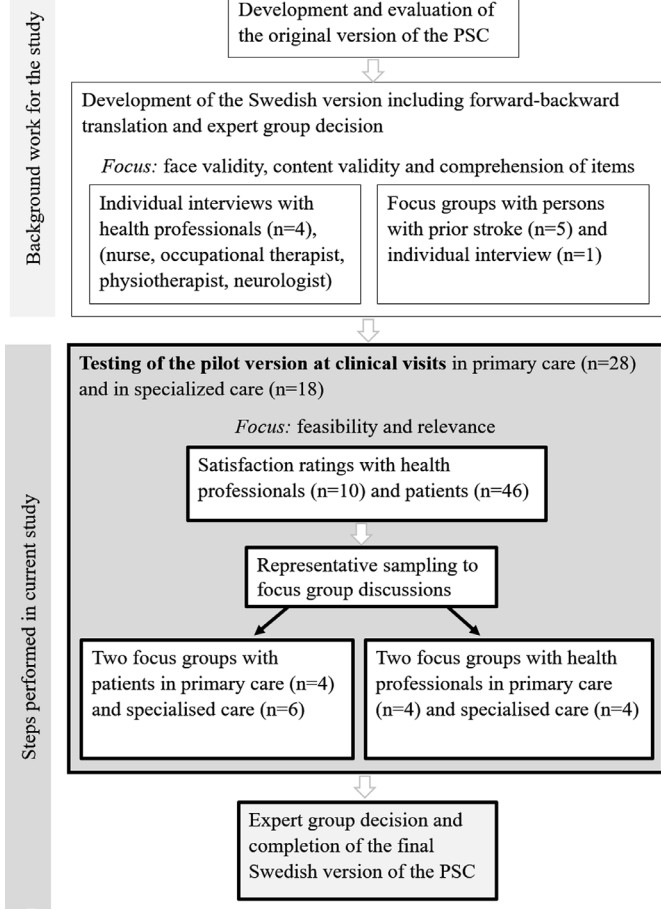

**Figure 1** The steps included in the validation and cross-cultural adaptation of the Swedish version of the poststroke checklist (PSC) including background work and current study.

and facilitate referrals.[11 12] Although the PSC has been found to be feasible and useful,[12–14] knowledge is lacking about the perspectives of patients and health professionals (HPs) regarding its use in the Swedish healthcare context. The aim of this study was to evaluate how people with stroke and HPs perceive the use of the PSC, with a focus on feasibility and relevance.

## METHODS
### Study design
The study has an exploratory design. To capture the feasibility of using the PSC, we combined it with a satisfaction questionnaire and focus group discussions,[15 16] in line with guidelines for complex interventions.[17] This study is part of a validation and cultural adaptation process[18] of the PSC in Sweden (figure 1). By combining data collection methods, we expected to gain a deeper understanding of using the PSC as a tool to structure follow-up. The underpinning methodology in the focus groups is based on social constructivism.[16] Consolidated criteria for reporting qualitative research (ie, [COREQ] guidelines)[19] were followed for reporting qualitative data.

### Patient and public involvement
This study explores how people with stroke experience the PSC. People from the Swedish Stroke Association (patient association) were involved in the translation process, the pilot testing of the interview guide and focus group discussions and have been given a presentation of preliminary results.

### Participants
Participants were consecutively recruited while at a clinical visit in primary care or stroke specialised outpatient care at a university hospital, February 2015 to October 2015. The inclusion criterion was having had a stroke, regardless of the time of onset. The number of patients included was in accordance with a previous study[12] and principles for cross-cultural adaptation suggesting approximately 40.[18] Patients were excluded if cognitive impairment or insufficient knowledge of Swedish would have made the response to the PSC items unreliable. HPs from different clinics were invited to participate and selected to represent different professions. A purposive sampling was used with the attempt to achieve heterogeneity and homogeneity in the focus groups.[15 16] Written informed consent was obtained from all participants.

### Data collection
The PSC[11] has 11 items and is intended to identify poststroke problems. It was developed by a multiprofessional group of stroke experts, according to a Delphi process. Problem areas were chosen for having the greatest impact on patient quality of life that could be addressed with evidence-based interventions. The PSC includes secondary prevention, activities of daily living (ADLs), mobility, spasticity, pain, incontinence, communication, mood, cognition, life after stroke and relationship with family. A response scale includes 'yes' and 'no' and recommended referrals adjacent to each problem area.[11]

Data collection was conducted in outpatient clinical facilities in two steps. For step 1, the HP administered the PSC to patients at a regular clinical visit (proxy responses were allowed). No additional training was given beyond the general instructions on the PSC. Participants were asked to reflect on the usefulness of the PSC as the basis for focus group discussions. Patients and HPs assessed satisfaction with the PSC after each visit through a satisfaction questionnaire with questions analogous to those used in a previous study.[12] The answers were rated on a Likert scale of 1–5, where 1 indicated not satisfied and 5 completely satisfied. To ensure anonymity, patient responses were collected in an envelope. Demographic data (time since index stroke, age, sex), time to administration of the PSC and HP profession were registered. In addition, if any referrals were made it was registered as yes/no for each patient without specification of what kind of referrals or standardised 'actions' to be taken. Of the patients who gave informed consent, additional patient characteristics (such as type of stroke, aphasia,

ADL dependency) were collected retrospectively from their medical records.

For step 2, the staff invited all participants during the follow-up to join in a focus group discussion. A set time of approximately 1–2 months was given between the visit and the focus group discussions. The first author (EK) telephoned participants willing to participate and sent them the study information letter, time for appointment and a copy of the PSC. In total, four focus groups were conducted using a question guide.[15] Each focus group met once for approximately 1.5 hours. The meeting was recorded and transcribed verbatim. Initially in the focus groups, the importance of bringing up different opinions was emphasised. At the end, an oral overview was presented to ensure that participant contributions were as they intended.

## Data analysis

Data gathered from PSC items, questionnaires and demographic information were analysed using descriptive statistics with Statistical Package for the Social Sciences V.24. Data gathered from focus groups were analysed following the analysis guidelines presented by Kreuger.[15] The aim of the analysis process was to describe the participants' perceptions and experiences based on the aim of the study.

Immediately after each focus group discussion, a written summary was created. The transcripts were coded using the computer software NVivo. First, the transcripts were read in their entirety to allow familiarity with the content as a whole, and discussions relevant to the aim were identified. Second, transcripts were systematised into categories based on similarities and differences in the discussions. Third, a descriptive summary was made for each category, adhering as closely as possible to the content of the raw data. Finally, these summaries in combination with selected quotes served as the basis for the interpretation and presented a deeper insight into the findings. In the analysis process, identified patterns were compared and contrasted across all four groups, resulting in an overarching thematic structure (for examples of the coding tree and themes see table 1). Quotations that showed the ongoing discussions[16] were selected to illuminate the

results. Based on sampling strategies[20] and when similar discussions recurred in all groups,[15] data gathering stopped.

The first author (PhD student, OT, woman) was the moderator and performed most of the analysis. Multiple coding, continuous interpretation of data and discussion of the emerging themes were completed together with the second author (PhD, OT, woman) to ensure accuracy of the analysis. The first author had knowledge about the study topic and the second author in qualitative methods. Both have conducted interviews previously. The third author (PhD, MD, woman) and last author (PhD, OT, woman) contributed with knowledge concerning revising and refining the themes. All authors have at least 20 years of experience in stroke rehabilitation.

## RESULTS

### Study group

The PSC was used in connection with a clinical visit in 46 patients. All patients lived in their homes. Most of them (65%) had experienced a stroke within 3 months at enrolment. The median time for hospitalisation was 8 days. Stroke severity at stroke onset was mild, with a median of 2 according to National Institutes of Health Stroke Scale. Table 2 shows the characteristics of the participants and the focus groups.

### Feasibility of the PSC

Forty patients (87%) had one or more problems identified by the PSC (figure 2). 'Life after stroke' was most common (61%), followed by cognition (56%) and mood changes (41%). A median of four problem areas per patient (range 0–9; IQR 1–5) was identified; the median in specialised care was 3 (IQR 1–5), and in primary care, it was 4 (IQR 1–5). Only six (13%) patients reported no problems. Most patients (70%) acknowledged having received information about secondary prevention. Referrals were registered in eight cases, slightly more often in primary care (n=6) than in specialised care (n=2). The time taken to

| Table 1 | Examples illustrating the coding tree | |
| --- | --- | --- |
| **Quote** | **Code** | **Subtheme** |
| P4: 'It has a lot to do with the competence of the person who's asking the questions so they can do the thinking to squeeze it all in.' (Group 1 patients) | The professionals' expertise and reasoning | The importance of HPs with stroke expertise and communication skills for capturing patient needs |
| P2: "So, I think it has just been positive, and it is also done so quickly". P1: 'That's also a positive'. P2: "Yeah, it's fast, but you can also develop it as much as you want. But asking the questions doesn't take too long". Moderator: 'Is it quick?' P1: "Yes. It also depends on what answers you get". P2: 'Yes'. (Group 4 HPs) | The administration of the PSC can be adapted, quickly done or more in-depth | The PSC supports continuity and referrals but depends on available resources |

HP, health professionals; PSC, poststroke checklist.

| Table 2 | Characteristics of patients and health professionals in the clinical outpatient visits and the focus group discussions | | |
|---|---|---|---|
| | Clinical visit (n=46) | Focus group 1 (n=4) | Focus group 2 (n=6) |
| Patients | | | |
| Primary care, rural | | x | |
| Specialised care, urban | | | x |
| Age (years) at inclusion | 70 (41–85) | 71 (58–78) | 74 (45–76) |
| Sex, male | 33, 72% | 4 | 5 |
| Education | | | |
| Mandatory | 20 | 1 | 4 |
| High school | 13 | 1 | 1 |
| University | 8 | 2 | 1 |
| Months since stroke | 3 (1–84) | 20 (3–84) | 3 (1–6) |
| Working at stroke onset (yes) | 13 | 2 | 1 |
| Length of hospitalisation, (days) | 8 (2–120) | 11 (5–82) | 8 (4–11) |
| History of stroke (yes) | 9 | 1 | 3 |
| Stroke characteristics | | | |
| Ischaemic/haemorrhagic | 36/5 | 4/0 | 4/2 |
| Right/left/posterior/bilateral | 19/16/5/2 | 3/1/0/0 | 3/2/1/0 |
| NIHSS | 2 (0–16) | 4 (3–10) | 2 (1–6) |
| Aphasia (yes) | 9 | 0 | 1 |
| Neglect (yes) | 4 | 1 | 0 |
| At discharge | | | |
| ADL independency (yes) | 34 | 3 | 6 |
| Wheel-chair use (yes) | 4 | 1 | 0 |
| | Clinical visit (n=10) | Focus group 3 (n=4) | Focus group 4 (n=4) |
| Health professionals | | | |
| Age (years) | 46 (35–63) | 43 (37–46) | 46 (35–55) |
| Primary care, rural | | | x |
| Specialised care, urban | | x | |
| Sex, male | 3, 30% | 0 | 1 |
| Nurse/OT/physician | 4/1/5 | 3/0/1 | 0/1/3 |
| Stroke experience (years) | | | |
| ≤5/5–10/10 | 2/2/6 | 0/1/3 | 2/1/1 |

Data are presented as number of persons (n) or median and range. Missing data from medical records (n=4).
ADL, activities of daily living; NIHSS, National Institutes of Health Stroke Scale; OT, occupational therapist.

administer the PSC was ≤15 min for 52%, ≤30 min for 43% and ≥45 min for 5%.

Four focus groups were conducted, and their characteristics are shown in table 2. One woman dropped out because of a scheduled medical examination. The focus group discussions revealed that the PSC structure in combination with room for dialogue could support egalitarian follow-up and identification of needs. A main theme and three organisational themes emerged in these discussions (figure 3).

### The content and relevance of the PSC
*Item relevance*
The items included in the PSC were considered relevant to all groups. Because stroke affects persons differently,

participants found it valuable that the PSC covers a broad spectrum of problems, although not all problems are relevant to every person with stroke.

*The PSC ensures coverage of important areas, but excluded areas could be missed*
Both patients and HPs stated that some issues might be overlooked if not specifically stated in the PSC such as nutrition, sexuality, vision, irritability and driving. The HPs discussed the appropriate amount of problem areas in the PSC. They wanted more areas yet preferred the checklist to be short and complemented by profession-specific assessments when needed. Some participants appreciated the recurring phrase 'since your stroke', but others preferred to hear/say it once at the beginning of the visit.

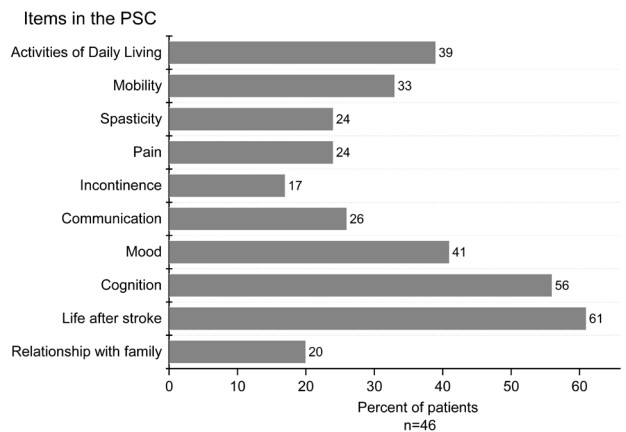

Items in the PSC

**Figure 2** Percentages of patients with identified problems in each poststroke checklist (PSC) item (n=46).

Patients perceived the PSC as easy to understand, while HPs expressed a concern about misunderstandings, especially for the items 'secondary prevention' and 'spasticity'.

### The PSC as a tool for dialogue

Patients and HPs both emphasised the need for dialogue to create mutual understanding. HPs described that knowledge and experience affected their ability to detect problems, while patients described differences in their ability to communicate problems.

#### Dialogue facilitates patients in expressing needs and engenders feelings of being cared for

Patients said that the PSC questions facilitated dialogue, leading to a greater likelihood that important areas would be elucidated and discussed. Memory problems, lack of initiative, fatigue or being less talkative were mentioned as barriers to dialogue that the PSC could address. The PSC gave clear direction for the structure of the dialogue and accordingly facilitating identification of problems. Nevertheless, using the PSC in combination with dialogue was

**Egalitarian follow-up through structure and dialogue**

| The content and relevance of the PSC | • Item relevance<br>• The PSC ensures coverage of important areas, but excluded areas could be missed |
| --- | --- |
| The PSC as a tool for dialogue | • Dialogue facilitates patients in expressing needs and engenders feelings of being cared for<br>• The importance of health professionals with stroke expertise and communication skills for capturing patient needs |
| The PSC as a systematic routine and a basis for egalitarian follow-up | • The PSC as a shared knowledge base to be individualised for each patient<br>• The PSC supports continuity and referrals but depends on available resources |

**Figure 3** Themes and subthemes derived from the focus group discussions with patients and health professionals regarding experiences of using the poststroke checklist (PSC).

seen as important. Patients stated that they might need time for consideration before answering the PSC questions, time that was often not given within the limits of the visits.

Generally, participants thought patients should have the opportunity to talk to a professional about stroke-related concerns and stated that the PSC could facilitate this exchange. Patients expressed that the PSC covers areas centring on them as a person, which made them feel cared for. The understanding of the HPs was that relatives often complemented information concerning problems that patients might neglect or forget to mention:

> P4: I think there is a great deal of importance to how much you are affected by the stroke. The more you are affected, the harder it is to think about the different facets of it (the areas in the PSC). (Strongly agreed on)
>
> (Group 1 patients)

#### The importance of HPs with stroke expertise and communication skills for capturing patient needs

To ensure that problems would be fully addressed, a professional's competence in stroke was seen as key by the participants. HPs feared that lack of HPs with stroke expertise might lead to problems going unrecognised. The PSC was seen as an asset as well as a barrier to dialogue. If too much focus was placed on posing the questions, participants experienced a decreased interaction. Sensitivity from the HPs and use of additional questions was seen as essential (eg, work issues, within the item 'life after stroke'):

> P7: No, there's only benefits, but it depends on how you use it, and if the staff think it is meaningful, so it's not just checked off. Rather that you have the opportunity to cover the things that each point is actually about.
>
> P8: Yeah, all the stuff that's crazy (difficulties after the stroke), follow-up that stuff.
>
> Moderator: That you have time to follow-up what is included in the point? Can you elaborate on that?
>
> P7: Yes, it's the topic this question is about, that you can elaborate on if you want to (…) the person with the checklist shouldn't be bound to it 100% and slavishly follow it, but understand signals from the patient and talk more broadly and connect it to the other things that depend on it. (Agreement)
>
> P9: Absolutely.
>
> P7: Otherwise, it just becomes mechanical; you can't be just like a computer asking questions.
>
> (Group 2 patients)

There were conflicting opinions about how to apply the questions in the PSC. Experienced HPs preferred to use it as a supplement for memory within a free dialogue. In contrast, inexperienced HPs perceived the specific

questions as good to assume and a basis for leading into other related concerns (eg, fatigue):

> P5: It's more about if you think that you should use standardised things for everyone, even for primary healthcare/outpatient care, I don't think that really works.

> P6: Exactly, that's the question, should you use the checklist just as it is, or should you use it for your own part and remember. That's the thing, because you can then approach each patient differently and get it all. But asking the exact same questions for each patient, I agree, that's really hard to do.

> (Group 3 HPs)

### PSC as a systematic routine and a basis for egalitarian follow-up
*The PSC as a shared knowledge base to be individualised for each patient*

The PSC was considered to increase knowledge about stroke and secure an egalitarian follow-up, especially for inexperienced HPs and patients with limited ability to express their needs. Even when the PSC was used, lack of HPs with stroke expertise and limited knowledge about opportunities for referrals were perceived as an obstacle to egalitarian follow-up. One suggestion was to add local referral opportunities and access to scientific references in conjunction with the PSC. Factors such as comorbidities and time since stroke must be taken into account because they could affect responses to PSC items.

*The PSC supports continuity and referrals but depends on available resources*

Participants addressed the need for regular follow-ups and considered the PSC to be a useful tool and basis for referrals. The use of the PSC was seen as a rapid way to cover the problem areas if only the questions were used, but when supplementary questions were needed, the time need also increased. One concern, especially among the physicians, was the time taken to administer the PSC in addition to their ordinary routines. HPs emphasised that the use of the PSC should be beneficial for the patients and in accordance with time limits and referral opportunities. Some considered it a bit rigid to go through all items if the patient experienced no problems, although it was observed that it could be completed quickly.

To enable preparation beforehand and make visits more time-efficient, a patient version of the PSC was proposed. Participants strongly emphasised that problems identified by the PSC should lead to appropriate intervention and not only an evaluation of current status:

> P11: The risk is you might get a false sense of security though. So, someone has asked the question, and I have answered 'yes' to this question; so I then expect something to happen. (Agreement from the others)

> P11: It's like, that's what decides the quality of what happens with the measures (…). It should end with me knowing how this information is taken and handled, what happens now. Not just that you do it and then that's great. (mumbles) Is it like statistics or what? (Group 1 patients)

Lack of opportunities for interventions as well as knowledge gaps were expressed by HPs as difficulties in meeting these expectations. A specific dimension of this problem was mentioned as leading to a risk of avoidance of discussing certain items.

By combining the results derived by different methods, additional aspects of the analysis can be demonstrated. The patients evaluated the satisfaction with the PSC as high (table 3). In addition, the focus group analyses gave insights into a wide range of factors that affect satisfaction, and its feasibility was exemplified by the importance of dialogue (figure 3). HP satisfaction with the PSC varied among patients (table 3). Participants perceived it as important to adapt the use of the PSC to individual aspects. Some differences regarding individual prerequisites are displayed in table 2, and others are mentioned in the qualitative analyses. HP stroke experience varied, especially in the primary care settings (table 2). In the focus group discussions, HPs with stroke expertise was perceived as important if subtle problems were to be properly recognised.

### DISCUSSION
The PSC is a relevant and feasible tool to identify patients who can benefit from targeted interventions, as noted by people with stroke and HPs. The original purpose of the PSC was to be an easy-to-use tool to detect poststroke

**Table 3** Evaluation of the use of poststroke checklist (PSC) based on satisfaction ratings (Likert 1–5) by patients and health professionals

| Satisfaction with: | Patients (n=46) median (IQR) | Health professionals (n=10) median (IQR) |
|---|---|---|
| Overall assessment where PSC was used | 5 (4–5) | 4 (3–4) |
| Identification of needs | 5 (4–5) | – |
| Identification of need (for each patient) | – | 3 (3–4) |
| Confidence in receiving support | 5 (4–5) | – |
| Guidance for referrals and treatment | – | 3 (2–4) |

problems, as well as a support for guiding referrals.[11] This study brings out an awareness about how follow-up through the PSC could be enhanced by user perceptions and suggested strategies. This knowledge could add important insights when implementing the PSC in line with the World Stroke Organisation recommendations. The focus group discussions raised issues concerning prerequisites when using the PSC. These include HP with stroke expertise, room for dialogue and how the identified needs were addressed.

The wide range of poststroke problems identified in the present study demonstrated the relevance of the PSC, with a median of four problems per patient. Of note, reported problems on specific PSC items differ considerably; 'life after stroke', cognition and continence vary when comparing among countries.[12–14] Comparison should be made with caution since the groups studied differ with respect to, for example, case-mix, sampling strategies and inclusion criteria in the studies. Furthermore, based on issues raised in the focus group discussions, likely causes of these discrepancies in reported problems could be HP stroke expertise, opportunity for a dialogue and time limits on the administration of the PSC. Comorbidities also could affect responses to the PSC items due to respondents not being able to consider whether the problems are stroke related or not. Nevertheless, the wide range of identified problems alongside participant perceptions in this study stresses the relevance of using the PSC in clinical practice. The long-term consequences after stroke emphasise the need for a comprehensive long-term follow-up with a multidomain approach.[1 3 21–23]

The present study provides a deeper understanding of how the PSC structure could support patients in expressing their needs. Problems with communication and comprehension are common after stroke,[5] which influences decision-making during follow-up. Participation in decision-making requires health literacy, that is, the ability to understand health information and a capacity to argue for one's needs in relation to appropriate interventions.[7] In this study, despite perceptions that the PSC questions were easy to understand, dialogue was found to be crucial. Participants raised concerns about problem areas that could be missed depending on how the PSC was used. Results from using the PSC in the UK and Singapore[12] indicate that several problem areas could be indirectly identified. Awareness of the complexity of need identification underscores the role of the HP when using the PSC. Even if no unmet need is reported, a person can still identify as living with residual impairments and perceived problems in engaging in activities.[24] Participants expressed that identification of needs could be enhanced if more time were allowed for consideration of these needs and for additional questions; another help would be the opportunity to fill in the PSC beforehand. Current findings stress HP with stroke expertise and the need to make space for dialogue when administering the PSC to support needs identification and decision-making.

The results from the present study highlight the dialogue between the patient and HP, which is central in healthcare.[9 10 25] Experienced HPs argued that they could cover most topics using open-ended questions. In contrast, others emphasised the value of articulating the PSC questions literally. To enable investigation of specific areas, closed questions can be important[26] and facilitate situations involving patients with communication difficulties. However, the PSC instructions do not hinder its use in a looser way as long as all areas are captured. The result shows that patients can benefit from a clear structure when the PSC is used. Participants in all focus groups agreed on the benefits of going through the areas in the PSC in a way that ensures identification of unmet needs. In addition, using the PSC in combination with dialogue supports the patients' capacity to communicate their needs. A narrative communication, along with signs of problems, gives the HP a foundation for planning care together with the patient and creates conditions for patients to make appropriate health decisions.[10]

The PSC can improve clinical pathways in healthcare by its structure and guidance for further referrals. Creating a plan to take care of identified needs and locally adapted pathways to support access to appropriate interventions, was noted in the focus group discussions as essential.

### Strengths and limitations

A strength of this study is that patients were partners throughout the project, from the translation process to participation in the focus group discussions. In addition, the mix of methods made it possible to explore the feasibility of the PSC at different levels compared with previous research. Few persons in this population had lived with their stroke for a long time, and only one of them had a severe stroke, which might have affected which problems were identified (eg, spasticity). This naturalistic design in an ordinary outpatient context, however, is representative of the Swedish population, where the majority have mild stroke.[27] The time (1–2 months) between the follow-up visit where the PSC was used and the focus group discussion could be a risk for recall bias. However, the focus group methodology enabled exploring a range of opinions of people across groups, and together, the participants contributed to rich discussions. Although the attempt was to obtain heterogeneity and homogeneity in the focus groups, the majority of HPs were women and the majority of the patients were male. Because the purposive sampling of HPs were made based on healthcare facilities already chosen and the defined time limit between the visit and the focus groups, the sex distribution were out of our influence. However, heterogeneity was achieved with respect to different outpatient settings, patients having a range of ages, stroke characteristics and education levels, and HPs with a range of professional roles and experience in stroke. To strengthen the transferability of the findings, a comprehensive description of the study context, participant characteristics, data collection and

analysis process are included in the 'Methods' section. Nevertheless, there are limitations regarding the transferability of the findings outside of the Swedish healthcare context. To ensure the feasibility of using the PSC in another context, a cross-cultural validation is needed. However, because the World Stroke Organisation recommends using the PSC globally, these results contribute to a deeper understanding of its feasibility that can also be useful to other countries.

## CONCLUSIONS

The PSC is a feasible and relevant tool to support egalitarian follow-up and identify patients who can benefit from targeted interventions after stroke. HPs stroke expertise, room for dialogue and caring for identified needs were raised as important issues to consider when using the PSC. Nutrition, sexuality, driving, work, vision, irritability and fatigue were areas mentioned that might need to be addressed within the discussions by HPs using the checklist. The PSC can facilitate patients in expressing their needs, enhancing their ability to participate in decision-making.

**Acknowledgements** The authors would like to thank the health professionals and patients for sharing their experiences. The authors would also like to thank Dr Kate Bramley-Moore for translation help regarding quotations from the transcribed interviews.

**Contributors** EKK, ÅL-N and KSS contributed to the design of the study. EKK conducted the interviews and analysed the data together with GC, involving KSS and ÅL-N in the final stages of the analysis. EKK wrote the first version of the manuscript, which was reviewed by GC, ÅL-N KSS. All four authors contributed to and approved the final manuscript.

**Funding** The study was funded in part by the Swedish Stroke Association; the Local Research and Development Board for Gothenburg and Södra Bohuslän; the Swedish Heart and Lung Foundation; the Swedish Brain Foundation; an unconditional grant from Allergan; the Hjalmar Svenssons Foundation; Neurological Foundation; Greta and Einar Askers Foundation; Rune and Ulla Amlövs Foundation; Per-Olof Ahls Foundation and John and Brit Wennerströms Foundation. The study was supported by grants from the Swedish state under the agreement between the Swedish government and the county councils, the ALF agreement (ALFGBG-719 80).

**Competing interests** None declared.

**Patient consent for publication** Not required.

**Ethics approval** The regional ethical review board in Gothenburg approved the study (no. 521–14).

**Provenance and peer review** Not commissioned; externally peer reviewed.

**Data sharing statement** Due to ethical restrictions, data are available on request. Researchers can submit requests for data to the authors (contact: ks.sunnerhagen@neuro.gu.se). Complete data from interviews cannot be made publicly available for ethical and legal reasons, according to the Swedish regulations http://www.epn.se/en/start/regulations/. Public availability would compromise participant privacy or confidentiality. On request, a list of condensed meaning units or codes can be made available after removal of information that may risk the confidentiality of the participants. To access data, please contact the first author: (emma.kjork@neuro.gu.se).

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
