## [Reviewer comments · BMJ Open]

ARTICLE DETAILS

TITLE (PROVISIONAL)	Experiences using the Post-Stroke Checklist in Sweden with a focus on feasibility and relevance: a mixed-method design
AUTHORS	Kjörk, Emma K; Carlsson, Gunnel; Sunnerhagen, Katharina; Lundgren-Nilsson, Asa

VERSION 1 – REVIEW

REVIEWER	Benjamin Hotter Charité Universitätsmedizin Berlin
REVIEW RETURNED	12-Dec-2018

GENERAL COMMENTS	The authors present an interesting study on the implementation and feasibility as well as qualitative analysis of the Post Stroke Checklist in a Swedish environment. The issue is of high importance in the field of stroke research and currently underrepresented in the literature. I would argue to publish this manuscript, although I would like to raise minor concerns: - While the authors let us know about the frequency of affected domains, we do not really learn anything about the conclusions drawn from the PSC for the individual patients. While they state that 8 referrals were registered, I would be interested in what kind of referrals they were and how many of these patients were referred. Furthermore, the methods section should state whether there was an agreed upon standardised "action" to be taken, once an item of the PSC was considered pathological.- I would recommend discussing the skewed gender representation (especially, but not only in the focus groups). Especially with the qualitative approach, it would have been of great interest to have a more diverse array of stakeholders to voice their opinion.
--

REVIEWER	Giorgio Sandrini Professor of Neurology, Dept. of brain and behavior Sciences, University of Pavia, Pavia (Italy)
REVIEW RETURNED	06-Jan-2019

GENERAL COMMENTS	PSC is extensively used in clinical practice in several countries. Even if its use is recommended by World Stroke Organization, advantages and limits of PSC are yet argument of debate. The paper gives an interesting contribution about this issue, but it needs a minor revision, since some points need to be better clarified. In particular: - the number of patients is limited and does not allow to compare different subgroups according to specific clinical
--

	items(i.e.spasticity).The authors are asked to better clarify that the results are preliminary and to define the next steps of their job  -a large bias in the timing of the visit can represent a relevant limit for instance in detecting some disorders(spasticity,in particular) -several HP are involved in the study but it is unclear if someone of them is particularly indicated to detect unmet needs -PSC can be useful in detecting some unmet needs ,but the authors have to better specify as to improve the clinical pathways for identifying them -co-morbidities can play a relevant role in post-stroke patients worsening their clinical condition and disability.This issue need to be more extensively discussed
--	--

REVIEWER	Dr Ian Wellwood Senior Research Associate University of Cambridge, UK
REVIEW RETURNED	15-Jan-2019

GENERAL COMMENTS	Thanks for submitting the manuscript which describes a study of utility / feasibility / relevance of the post stroke checklist (PSC) in the Swedish health system. Overall the authors have made considerable efforts to conduct and report the study and the methods and findings will be of interest to those working with mixed methods and needs assessment in clinical populations as well as those using, or considering using, the PSC. The local results are placed within a wider context and add information about the validity, cultural adaptation and utility of this tool that others can interpret. The comments below relate to the reviewers' checklist items above. 6. Are the outcomes clearly defined? In the Methods section there is not a specific focus on outcomes – but the work is described as exploratory and undertaken within a validation framework. Item 11. Are the discussion and conclusions justified by the results a) In the Discussion – the authors note that there are differences in PSC results between samples in different countries reported in the literature – and then suggest “likely causes“ for these. There should be further consideration of other likely causes of differences in needs (e.g. case-mix, sampling frames and inclusion criteria), when comparing these samples / populations and a note of caution when attempting to compare across studies in this manner. Minor points: b) In the Strengths and limitations section, please reference the statement about the majority of strokes being “mild“ in Sweden. c) In the Conclusions section I would find it helpful to note, and perhaps repeat, some of the examples of clinically relevant information (nutrition, sexuality, driving, fatigue) that is not explicitly covered by the items in the PSC – and that might need to be picked up in discussions by health professionals using the checklist. Item 12. Are the study limitations discussed adequately? a) It would be helpful to note and comment on the proportion of males in the patient sample and females in the health professionals group. Percentages might usefully be added for these items in Table 2. This is worth noting as the authors
--

	describe purposive sampling to obtain “heterogeneity and homogeneity” and also comment on how representative the population might be in the Strengths and Limitations sections. b) In the final sentence the authors discuss their contribution to the understanding of the feasibility of using the PSC - however I think more emphasis needs to be made on the limitations of transferability of this information outside of the Swedish healthcare context. (The authors themselves note this point about the specific context of their study in the Introduction section). Item 15. Is the standard of written English acceptable for publication? Overall this is acceptable and the authors sought editorial and translation advice. I have suggested below a small number of amendments based on English useage that the authors / editor may wish to consider.  • The use of “people“ is considered more popular than “persons“. (point 3 in Strengths & Limitations section).  • While I think I understand the concept of “stroke competence“, this could be defined or clarified – might it be re-phrased in the paper as for example “HP experienced in stroke care“ or “HP with stroke expertise“ or similar? • Minor typographical: Page 11 – “i.g“ = “e.g.“ Table 1 under“Code“ – professionals (apostrophe required)
--	---

VERSION 1 – AUTHOR RESPONSE

Comments for the reviewers

The changes that are made in the manuscript are yellow marked and the page and line numbers are given, in this document. Comments regarding the changes are written in blue text.

Reviewer: 1

Reviewer Name: Benjamin Hotter

Institution and Country: Charité Universitätsmedizin Berlin Please state any competing interests or state ‘None declared’: None declared

Please leave your comments for the authors below The authors present an interesting study on the implementation and feasibility as well as qualitative analysis of the Post Stroke Checklist in a Swedish environment. The issue is of high importance in the field of stroke research and currently underrepresented in the literature. I would argue to publish this manuscript, although I would like to raise minor concerns:

- While the authors let us know about the frequency of affected domains, we do not really learn anything about the conclusions drawn from the PSC for the individual patients.

AUTHORS’ RESPONSE: Thanks for your comment. Since this study is conducted within a validation framework, we did not aim to focus on how the identified post-stroke problems affected individual patients or potential explanations for the PSC results. Due to the amount of data in the study, we had to prioritize the focus of the discussion towards the validation and utility aspects. However, it sure is an important area for further research.

While they state that 8 referrals were registered, I would be interested in what kind of referrals they

were and how many of these patients were referred. Furthermore, the methods section should state whether there was an agreed upon standardised "action" to be taken, once an item of the PSC was considered pathological.

AUTHORS' RESPONSE: Thanks for your comment. We do agree with you. It should have been valuable if we had specified the referrals and standardized actions before data collection. However, we didn't and that is a limitation of the study. Prerequisites for referrals and actions were different at the different study sites, e.g. in specialised care the nurses could discuss with the physician on daily basis and most certainly, they did not report that as a referral.

We have made changes in the method section as follows:

P.5, line 159-160 "In addition, if any referrals were made it was registered as yes/no for each patient without specification of what kind of referrals or standardised "actions" to be taken".

- I would recommend discussing the skewed gender representation (especially, but not only in the focus groups). Especially with the qualitative approach, it would have been of great interest to have a more diverse array of stakeholders to voice their opinion.

AUTHORS' RESPONSE: Thanks for your comment. We do agree with you that further discussions are needed. There are some explanations for the skewed gender distribution in the sample. In Sweden, women constitutes 50% of the stroke population but the mean age at stroke onset is 4 years older and is more likely to have comorbidities. Accordingly, women are more often referred to nursing homes and excluded from long-term follow-up. In addition, the majority of professionals in health care in Sweden are women.

Changes have been done in the methods section, table 2 and in the strength and in the limitation section:

p.5, line 137. "...with the attempt to achieve heterogeneity and homogeneity.."

p. 15, line 433-438 "Although the attempt was to obtain heterogeneity and homogeneity in the focus groups the majority of HPs were women and the majority of the patients were male. The sex distribution were out of our influence since the purposive sampling of HPs were made based on health care facilities already chosen and the defined time limit between the visit and the focus groups were set beforehand."

p. 8, line 209. Table 2. Patients: Sex, male 33, 72%

HPs: Sex, male 3, 30%

Reviewer: 2

Reviewer Name: Giorgio Sandrini

Institution and Country: Professor of Neurology, Dept. of brain and behavior Sciences, University of Pavia, Pavia (Italy) Please state any competing interests or state 'None declared': none declared

Please leave your comments for the authors below PSC is extensively used in clinical practice in several countries. Even if its use is recommended by World Stroke Organization, advantages and limits of PSC are yet argument of debate. The paper gives an interesting contribution about this issue, but it needs a minor revision, since some points need to be better clarified.

In particular:

-the number of patients is limited and does not allow to compare different subgroups according to specific clinical items (i.e. spasticity).

The authors are asked to better clarify that the results are preliminary and to define the next steps of their job - a large bias in the timing of the visit can represent a relevant limit for instance in detecting some disorders (spasticity, in particular)

AUTHORS' RESPONSE: Thanks for your comment. We do agree with you that the PSC results should be interpreted with caution, since the possibility to detect problems can differ for obvious reasons e.g. relation between time since stroke and spasticity. We hope that table 2 gives the reader enough information about the patient characteristics in this sample to understand the context for this validation. We have previously shown (Opheim A et al, Sunnerhagen et al) that signs of spasticity can often be noted within the first 4 weeks after stroke but may vary with time (in some patients worsen and in some disappear). Therefore, the checklist is a base to identify spasticity but not sufficient to follow the course. However, this is not part of the present study.

In the present study, which is part of a validation process, it was not an aim to interpret the results of the PSC per se. Of course, it would be of greatest interest to conduct studies with the PSC as

outcome in larger samples with different stroke sub groups to be able to compare groups and detect differences between stroke subgroups. Next step in our process is conducting a study exploring if the PSC can detect problems in people with stroke living in nursing homes. This validation process with respect to Swedish conditions is a first step in enabling further studies with PSC as outcome.

We made changes in the discussion part as follows:

p. 14, line 378-380. "Comparison should be made with caution since the groups studied differs with respect to e.g. case-mix, sampling strategies and inclusion criteria in the studies".

p. 15, line 428. "...affected which problems were identified (e.g. spasticity)".

-several HP are involved in the study but it is unclear if someone of them is particularly indicated to detect unmet needs

AUTHORS' RESPONSE: Thanks for your comment. Most certainly, there are differences among HPs regarding their ability to detect unmet needs. The results in this study highlights the importance of the dialogue when using the PSC. One could assume that HPs with stroke expertise are more skilled in detecting unmet needs through a dialogue. However, this limited sample with present study design does not allow to draw any conclusions regarding different HPs ability to detect unmet needs. In the validation process, we included both generalists and specialists to capture opinions from HPs with different stroke experience. In table 2 years of stroke experiences is presented for the whole HP group and for each focus group separately.

-PSC can be useful in detecting some unmet needs, but the authors have to better specify as to improve the clinical pathways for identifying them

AUTHORS RESPONSE: Thank you for mention this aspect. We agree with you that one important advantage with the PSC is through enabling improvements of the clinical pathways for identifying unmet needs.

We made a change in the discussion section:

p. 15, line 417-420. "The PSC can improve clinical pathways in health care by its structure and guidance for further referrals. Creating a plan to take care of identified needs and locally adapted pathways to support access to appropriate interventions, was noted in the focus group discussions as essential."

-co-morbidities can play a relevant role in post-stroke patients worsening their clinical condition and disability. This issue need to be more extensively discussed

AUTHOR RESPONSE: Thank you for your comment. We do fully agree with you regarding the impact of comorbidities in this population. We have added a sentence regarding comorbidities in the discussion:

p.14, 382-384. "Comorbidities also could affect responses to the PSC items due to respondents not being able to consider whether the problems are stroke related or not."

Reviewer: 3

Reviewer Name: Dr Ian Wellwood

Institution and Country: Senior Research Associate University of Cambridge UK Please state any competing interests or state 'None declared': None declared

Please leave your comments for the authors below Thanks for submitting the manuscript which describes a study of utility / feasibility / relevance of the post stroke checklist (PSC) in the Swedish health system.

Overall the authors have made considerable efforts to conduct and report the study and the methods and findings will be of interest to those working with mixed methods and needs assessment in clinical populations as well as those using, or considering using, the PSC. The local results are placed within a wider context and add information about the validity, cultural adaptation and utility of this tool that others can interpret.

The comments below relate to the reviewers' checklist items above.

6. Are the outcomes clearly defined?

In the Methods section there is not a specific focus on outcomes – but the work is described as exploratory and undertaken within a validation framework.

AUTHORS' RESPONSE: As you describe, this study focus on validity and utility of the PSC rather than the individual patient outcome on the PSC items.

Item 11. Are the discussion and conclusions justified by the results

a) In the Discussion – the authors note that there are differences in PSC results between samples in different countries reported in the literature – and then suggest “likely causes“ for these. There should be further consideration of other likely causes of differences in needs (e.g. case-mix, sampling frames and inclusion criteria), when comparing these samples / populations and a note of caution when attempting to compare across studies in this manner.

AUTHORS' RESPONSE: Thanks for your comment. We do fully agree with you and have changed the manuscript accordingly.

p. 14, line 378-380. “Comparison should be made with caution since the groups studied differs with respect to e.g. case-mix, sampling strategies and inclusion criteria in the studies.”

Minor points:

b) In the Strengths and limitations section, please reference the statement about the majority of strokes being “mild“ in Sweden.

AUTHORS' RESPONSE: We have added a reference relevant for our area in Sweden.

...where the majority have mild stroke.²⁷

c) In the Conclusions section I would find it helpful to note, and perhaps repeat, some of the examples of clinically relevant information (nutrition, sexuality, driving, fatigue) that is not explicitly covered by the items in the PSC – and that might need to be picked up in discussions by health professionals using the checklist.

AUTHORS' RESPONSE: Thanks for your suggestion. We have made changes in the abstract and in the conclusion section:

(Due to word limits the abstract text had to be shortened):

p. 2, line 62-63 “Nutrition, sexuality and fatigue were areas mentioned that might need to be addressed within the discussions”.

and p. 16, line 452-453. “Nutrition, sexuality, driving, work and fatigue were areas mentioned that might need to be addressed within the discussions by HPs using the checklist”.

Item 12. Are the study limitations discussed adequately?

a) It would be helpful to note and comment on the proportion of males in the patient sample and females in the health professionals group. Percentages might usefully be added for these items in Table 2. This is worth noting as the authors describe purposive sampling to obtain “heterogeneity and homogeneity“ and also comment on how representative the population might be in the Strengths and Limitations sections.

AUTHORS' RESPONSE: Thanks for your comments and suggestions. We do agree that the representativeness of the population needs to be further discussed. There are some explanations for the skewed gender distribution in the sample. In Sweden, women constitutes 50% of the stroke population but the mean age at stroke onset is 4 years older and is more likely to have comorbidities. Accordingly, women are more often referred to nursing homes and excluded from long-term follow-up. In addition, the majority of professionals in health care in Sweden are women.

Changes have been done in the methods and in the strength and limitation section:

p.5, line 137. “...with the attempt to achieve heterogeneity and homogeneity..”

p. 15, line 433-438. “Although the attempt was to obtain heterogeneity and homogeneity in the focus groups, the majority of HPs were women and the majority of the patients were male. Because the purposive sampling of HPs were made based on health care facilities already chosen, the defined time limit between the visit and the focus groups, the sex distribution were out of our influence. However, heterogeneity was achieved with respect to..”

We have made changes in the table 2 as follows:

p. 8, line 209. Table 2. Patients: Sex, male 33, 72%

HPs: Sex, male 3, 30%

b) In the final sentence the authors discuss their contribution to the understanding of the feasibility of using the PSC - however I think more emphasis needs to be made on the limitations of transferability of this information outside of the Swedish healthcare context. (The authors themselves note this point about the specific context of their study in the Introduction section).

AUTHORS' RESPONSE: Thank you for your input. We have discussed the transferability of the study further in the limitation section:

p.16, line 442-444. “Nevertheless, there are limitations regarding the transferability of the findings outside of the Swedish healthcare context. To ensure the feasibility of using the PSC in another context, a cross-cultural validation is needed”. “...can also be useful to other countries”.

Item 15. Is the standard of written English acceptable for publication?

Overall this is acceptable and the authors sought editorial and translation advice.

I have suggested below a small number of amendments based on English useage that the authors / editor may wish to consider.

- The use of “people“ is considered more popular than “persons“. (point 3 in Strengths & Limitations section).

AUTHORS' RESPONSE: *Thank you for your relevant suggestions. We have changed from “persons with” to “people with” throughout the manuscript.*

- While I think I understand the concept of “stroke competence“, this could be defined or clarified – might it be re-phrased in the paper as for example “HP experienced in stroke care“ or “HP with stroke expertise“ or similar?

AUTHORS' RESPONSE: *Thanks for your advice. We have changed the term “stroke competence” to “stroke expertise” throughout the manuscript:*

- Minor typographical:

Page 11 – “i.g“ = “e.g.“

AUTHORS' RESPONSE: *We have changed the text in the result section.*

p.11, line 303 (e.g. fatigue):

Table 1 under “Code“ – professionals (apostrophe required)

AUTHORS' RESPONSE: *Thanks for noticing, we have changed it accordingly in the table.*

p7, line 199. The professionals'

VERSION 2 – REVIEW

REVIEWER	Benjamin Hotter Charité Universitätsmedizin Berlin, Germany
REVIEW RETURNED	13-Feb-2019
GENERAL COMMENTS	The authors met my concerns with a more in depth discussion.
REVIEWER	Giorgio Sandrini Dept.Brain and Behavior Sciences,University of Pavia,Italy
REVIEW RETURNED	09-Feb-2019
GENERAL COMMENTS	The paper can be accepted for publication
REVIEWER	Dr Ian Wellwood Institute of Public Health University of Cambridge, UK
REVIEW RETURNED	15-Feb-2019
GENERAL COMMENTS	Thank you for resubmitting your manuscript. The changes that are suggested address the points raised by the reviewers and I agree with the authors that it is now a stronger piece of work for the readers. I had two minor points that the authors may wish to address in any final version.  • When noting some of the items not covered by PSP (but that might be covered in a discussion with HP) it may be worth noting

	(alongside nutrition, sexuality, driving, work and fatigue) that vision and irritability were also raised. These are mentioned in the text, but since they are likely to be clinically relevant, it would be good to see them listed with the other points.  • Minor typographical errors in the replaced text: P 16 Conclusion – “addressed” = “addressed” P 14 “..groups studied differs with respect..” - “differs” = “differ”
--	---

VERSION 2 – AUTHOR RESPONSE

Comments to reviewers'

Reviewer(s)' Comments to Author:

Reviewer: 2

Reviewer Name: Giorgio Sandrini

Institution and Country: Dept.Brain and Behavior Sciences,University of Pavia,Italy Please state any competing interests or state 'None declared': none declared

Please leave your comments for the authors below The paper can be accepted for publication

Reviewer: 1

Reviewer Name: Benjamin Hotter

Institution and Country: Charité Universitätsmedizin Berlin, Germany Please state any competing interests or state 'None declared': None declared

Please leave your comments for the authors below The authors met my concerns with a more in depth discussion.

Reviewer: 3

Reviewer Name: Dr Ian Wellwood

Institution and Country: Institute of Public Health University of Cambridge UK Please state any competing interests or state 'None declared': None declared

Please leave your comments for the authors below Thank you for resubmitting your manuscript. The changes that are suggested address the points raised by the reviewers and I agree with the authors that it is now a stronger piece of work for the readers.

I had two minor points that the authors may wish to address in any final version.

- When noting some of the items not covered by PSP (but that might be covered in a discussion with HP) it may be worth noting (alongside nutrition, sexuality, driving, work and fatigue) that vision and irritability were also raised. These are mentioned in the text, but since they are likely to be clinically relevant, it would be good to see them listed with the other points.

Comments to reviewer: Thanks for your suggestions. We agree with you and have made changes in the conclusion section.

P. 16 “Nutrition, sexuality, driving, work, vision, irritability and fatigue were areas mentioned...”

- Minor typographical errors in the replaced text:

P 16 Conclusion – “addressed” = “addressed”

P 14 “..groups studied differs with respect..” - “differs” = “differ”

Comments to reviewer: Thanks for noticing. We have made the suggested changes in the discussion and in the conclusion section as follows:

P. 16 “..need to be addressed within the discussions.”

P. 14 “...groups studied differ with respect to e.g. case...”